# Design and Finite Element Analysis of Artificial Braided Meniscus Model

**DOI:** 10.3390/ma16134775

**Published:** 2023-07-01

**Authors:** Jiakai Wei, Wuxiang Zhang, Xilun Ding

**Affiliations:** 1School of Mechanical Engineering and Automation, Beihang University, Beijing 100191, China; by1707111@buaa.edu.cn (J.W.); xlding@buaa.edu.cn (X.D.); 2Ningbo Institute of Technology, Beihang University, Ningbo 315832, China

**Keywords:** meniscus prostheses, the three-dimensional braiding, finite element analysis, elastic properties

## Abstract

Currently, artificial meniscus prostheses are mostly homogenous, low strength, and difficult to mimic the distribution of internal fibers in the native meniscus. To promote the overall mechanical performance of meniscus prostheses, this paper designed a new artificial braided meniscus model and conducted finite element analysis. Firstly, we designed the spatial fiber interweaving structure of meniscus model to mimic the internal fiber distribution of the native meniscus. Secondly, we provided the detailed braiding steps and forming process principles based on the weaving structure. Thirdly, we adopted the models of the fiber-embedded matrix and multi-scale methods separately for finite element analysis to achieve the reliable elastic properties. Meanwhile, we compared the results for two models, which are basically consistent, and verified the accuracy of analysis. Finally, we conducted the comparative simulation analysis of the meniscus model and the pure matrix meniscus model based on the solved elastic constants through Abaqus, which indicated a 60% increase in strength.

## 1. Introduction

The meniscus is one of the crucial structures that form the knee joint, which has significant functions such as bearing loads, buffering loads, stabilizing joints, and lubricating joints [1]. Meanwhile, it also protects joint cartilage and possesses considerable physiological effects. Once the meniscus is damaged, it is difficult to self-repair. Besides, the current treatment methods and repair materials are severely limited. The general treatment methods include suturing, collagen implantation, meniscus replacement, etc. [2,3]. The better treatment method for patients with severe injury is performing meniscus transplantation surgery. However, allograft meniscus transplantation possesses potential risks, such as rejection and disease transmission, and it is seriously limited by the lack of donor sources. Therefore, the preparation of artificial meniscus prostheses has attracted increasingly attention [4]. The meniscus is a semi-lunar-shaped fibrocartilage disc with a triangular section, which is covered with a thin layer of fibrocartilage on the surface and has dense collagen fibers mixed with abundant elastic fibers inside. The meniscus has the thin and dense medial edge and smooth surface, which is composed of fibers and hyaline cartilage and is abrasion-resistant and impact-resistant. Nevertheless, the lateral edge of the meniscus is wide, loose, and elastic and suitable for blood vessels to grow in, which is composed of cartilage and relatively thick annular fibers to provide the capacity fixed to the articular capsule [5,6]. Currently, the collagen meniscal implant is wildly applied in clinic, which is a 3D reticular formation composed of type I collagen that originated from bovine Achilles tendon and is beneficial for cellular ingrowth due to the abundance of glycosaminoglycans [7]. The CMI has the superiority of biocompatible, bioresorbable, and regenerative properties, while is limited in short usage period [8]. The meniscal scaffold Actifit is biodegradable and exceedingly porous, which contains two components: 80% flexible material polycaprolactone and 20% stiff material PU [9,10,11]. Actifit has superior biomechanical properties and surgical handling compared to CMI [12]. Besides, polyurethane meniscus stent, poly(vinyl alcohol hydrogel) (PVA-H) meniscus, silk meniscus, and other matrices are mostly homogeneous materials, which are impossible to mimic the complex structure of physiological meniscus, such as longitudinal fibers and circumferential fibers [13,14,15]. The meniscal implant NUsurface is composed of a polycarbonate-urethane (PCU) matrix reinforced by embedded polyethylene fibers, which can provide long-term treatment and alleviated pain for the patients of meniscectomy. However, the distribution of the fiber is merely circumferential [16]. Generally, the existing artificial meniscus prostheses can still not completely mimic the structure of meniscus. Therefore, this paper proposed a 3D-brading meniscal structure to mimic the native meniscus.

To promote the overall mechanical performance of meniscus prostheses, this paper designed a new artificial braided meniscus model and conducted the finite element analysis to predict the elastic constants and the amount of strength improvement. For better biomimetic preparation of meniscus, continuous fiber-printing technology and weaving technology are integrated. Continuous fiber-reinforced polymer composites have advantages such as high specific strength and modulus, low density, and fatigue resistance. In particular, continuous fiber-reinforced thermoplastic composites (CFRTPCs) have received increasing attention in recent years due to their high impact toughness, short processing cycles, and good recyclability [17]. Matsuzaki et al. used the CFRTPC molding process based on melt deposition technology to prepare continuous carbon fiber-reinforced PLA composites, which showed a 5-fold and 3.4-fold increase in tensile modulus and strength compared to pure poly(lactic acid) (PLA) samples, respectively [18]; Tian et al. used carbon fiber as the reinforcement phase and ABS as the matrix to study its reinforcement mechanism and performance. The bending strength and modulus of printed composite material samples are approximately 127 MPa and 7.72 GPa, which is almost 6 times that of ABS samples prepared by traditional Fused Deposition Modeling (FDM) process and 3 times that of injection molded samples [19]. However, there are still serious problems in the extrusion process of continuous fiber-reinforced composite materials: the anisotropy of composite components is severe, mainly manifested as poor interlayer and interlayer bonding [20,21]. Fortunately, the three-dimensional weaving technology can incorporate the Z-fiber into the prosthesis, which bind layers tightly together. For the parts of the finite element analysis, benefiting from the rapid development of Abaqus software, the represent volume element (RVE) of fiber-embedded matrix and multi-scale methods are increasingly adopted to predict the elastic constants. Zhang et al. applied a unit cell to analysis the transverse impact behaviors of the 3D braided composites composed of carbon and epoxy through the finite element analysis method [22]. Zhang D et al. predicted the elastic properties of 3D braided composites through the fiber-embedded matrix method [23]. Chen et al. applied a finite multiphase element approach to predict the elastic constants of 3D braided composites [24]. Cox et al. proposed a binary model to predict the mechanical properties of 3D braided composites [25].

The present work is aimed at designing a new artificial braided meniscus model and predicted the elastic constants through the finite element analysis. The specific workflow is shown as follows: First, the spatial fiber interweaving structure of the meniscus model is designed to mimic the internal fiber distribution of the native meniscus. Then, the detailed braiding steps and forming process principles are provided based on the weaving structure and braiding theory. Subsequently, the models of fiber-embedded matrix and multi-scale methods are separately adopted for finite element analysis to achieve the reliable elastic properties. Meanwhile, the calculated results of two models are compared, which indicated the basically consistence and verified the accuracy of analysis. Finally, the comparative simulation analysis of the meniscus model and the pure matrix meniscus model are conducted based on the solved elastic constants through Abaqus.

## 2. Design of the Meniscus’s 3D Braiding Structure and Technology

### 2.1. The Structure of Designed 3D Braided Meniscus

The meniscus shown in Figure 1a is a half-moon-shaped fibrocartilage structure situated between the femoral condyle and the tibial plateau of the knee joint, one locates on the inside and another on the outside, and the shape of the lateral meniscus is more round, as shown in the Figure 1b. The main ingredients are 72% body fluid (water), 22% collagen, 0.12% DNA, and 0.8% total polysaccharide [26]. The highest proportion of collagen is type I collagen, and type II, III, V, and VI collagen also exist. The distribution of collagen fibers is divided into three layers as shown in Figure 1b: the shallow layer, where the fibers are distributed along the radial direction, and the shear and tear resistance are achieved; the middle layer, where the fibers are distributed along the parallel or circumferential direction to resist the circumferential stress during load bearing; and the deep layer, where fiber bundles are distributed along the edge of the parallel meniscus [27,28,29]. The size of the medial meniscus is obviously different from that of the lateral meniscus: the length of the lateral meniscus is about 32.4~35.7 mm, and the width is about 26.6~29.3 mm; the length of the medial meniscus is 40.5~45.5 mm, and the width is 27 mm [30,31].

Therefore, the three-dimensional braided structure of the meniscus is designed as shown in the Figure 1a, the overall length, width, and height of which is 32 mm, 26 mm, and 11 mm, respectively. In order to better simulate the wedge-shaped shape of the meniscus, the truss structure is divided into five layers in the height direction, and the woven fibers are arranged in a gradual decreasing manner. The parts from the bottom to the top are divided into one, two, and three levels. The first and second levels each contain the same two layers, and the third level constitutes one layer. The first level is shown in Figure 2e, with six transverse fibers distributed in a half-moon shape. The transverse fibers have a length of about 45 mm, a width of about 2 mm, a height of about 1 mm, and an interval between the fibers of about 2 mm; there are 13 longitudinal fibers that diverge from the inside to the outside at a certain angle, mimicking the characteristics of physiological meniscus where the fibers are closely spaced inside and sparsely spaced outside. The longitudinal fibers have a length of about 26 mm, a width of about 2 mm, a height of about 1 mm, and an angle between the fibers of about 15 degrees. The second level is shown in Figure 2f, has 4 transverse fibers with length of about 40 mm and 12 longitudinal fibers shortened by 8 mm compared to the first stage. The third level is shown in Figure 2g, which has two transverse fibers with length of about 35 mm and eight longitudinal fibers with a length shortened by 6 mm compared to the second level. The white fibers in Figure 2c,d connect the interlaminar fibers in the braiding structure to form a 3D interlocking structure, mimicking the radial fibers of the meniscus; meanwhile, they compensate for the lack of interlayer bonding and elevate the overall stability and damage resistance of the braiding structure.

### 2.2. 3D Braiding Meniscus’s Technology

The integrated molding fabric produced by traditional 3D braiding technology has a single internal fiber distribution, which is difficult to satisfy the personalized distribution requirements of meniscus’s internal fibers. Therefore, the 3D braiding meniscus’s structure is manufactured by combining continuous fiber 3D printing technology and 3D braiding technology. A detailed explanation of the braiding process steps is shown in Figure 3. First, based on the continuous fiber 3D printing technology, a truss structure as shown in Figure 3b is gradually printed and stacked according to the layered design structure of the meniscus, which provides support for subsequent fiber braiding. Second, adapting the technology of the hook and pick mechanism in the traditional braiding process to form a collar, the fiber is woven into the truss structure to form an interlocking braiding structure, as shown in Figure 3c. Third, the woven meniscus structure is injected and filled with suitable substrate in the corresponding mold to accomplish the preparation of integral 3D braiding meniscus prosthesis, as shown in Figure 3d.

### 2.3. The Braiding Steps

Figure 4 shows the operation steps of the braiding mechanism in detail.

Step 1: The end of thread-taking-up mechanism runs to the bottom of the corresponding truss cell, and the crochet runs to the adjacent truss cell. Meanwhile, the lead and the end of the crochet keep an appropriate distance in the vertical direction.

Step 2: The rotation of the reel in the rope driving mechanism drives the rotating shaft in the lead pin to rotate; then, the end of the lead pin guides the braided fiber to the bottom of the adjacent cell. The hook mechanism moves back and forth in a straight line from top to bottom to hook the thread ring and bring it back to the corresponding position in Step 1.

Step 3: The winding shaft of the rope driving mechanism rotates reversely to restore the lead pin to the corresponding position in Step 1. Meanwhile, the lead pin and crochet run to the corresponding position of the truss cell of next row of. At this time, the thread loop formed by the crochet in Step 2 is carried from the cell in the previous row to the cell in the next row, during which the thread loop slides onto the crochet rod. The lead wire is also introduced from the cell in the previous row to the bottom of the cell in the next row.

Step 4: Repeat the operation of the mechanism in Step 2. Meanwhile, the new thread ring hooked by the crochet will be sleeved into the thread ring formed in Step 2. The thread-taking-up mechanism drives the lead wire to tighten in the upward process of the crochet to separate the thread ring formed in Step 2 from the crochet. Circulating four steps to complete the integral braiding.

## 3. Analytical Method

### 3.1. Materials

Polycarbonate polyurethane (PCU) was selected as the matrix due to its excellent mechanical properties, thermoplasticity, and high porosity, which facilitates the preparation of artificial meniscus and has brilliant biological adaptability [32,33,34,35,36]. Kevlar was employed as the reinforced fiber, which has the excellent performance of high strength, low density, toughness, corrosion resistance, and fatigue resistance [37,38,39,40,41]. The detailed mechanical properties of the selected materials are shown in Table 1.

### 3.2. The Method of Fiber-Embedded Matrix

The matrix material is isotropic, the normal phase vector of its compliance matrix is ${\left[s_{ij}^{m}\right]}_{\delta }$, the tangential vector is ${\left[s_{ij}^{m}\right]}_{\tau }$, and the flexibility matrix is $\left[s_{ij}^{m}\right]$:(1)$${\left[s_{ij}^{m}\right]}_{\delta }=\left[\begin{array}{ccc} \frac{1}{E} & -\frac{v}{E} & -\frac{v}{E}\\ & \frac{1}{E} & -\frac{v}{E}\\ \mathrm{symmetry} & & \frac{1}{E} \end{array}\right]{\left[s_{ij}^{m}\right]}_{\tau }=\left[\begin{array}{ccc} \frac{1}{G} & 0 & 0\\ & \frac{1}{G} & 0\\ \mathrm{symmetry} & & \frac{1}{G} \end{array}\right]\left[s_{ij}^{m}\right]=\left[\begin{array}{cc} {\left[s_{ij}^{m}\right]}_{\delta } & 0\\ 0 & {\left[s_{ij}^{m}\right]}_{\tau } \end{array}\right]$$

The relationship between shear modulus, Young’s modulus, and Poisson’s ratio is:(2)G=0.5E/(1+v)

The fiber is transversely isotropic, the normal phase vector of its compliance matrix is ${\left[s_{ij}^{f}\right]}_{\delta }$, the tangential vector is ${\left[s_{ij}^{f}\right]}_{\tau }$, and the flexibility matrix is $\left[s_{ij}^{f}\right]$.

$E_{11}$, $v_{12}\;\mathrm{and}\;G_{12}$ represent axial elastic modulus, axial Poisson’s ratio and axial shear modulus respectively; $E_{22}$, $v_{23}\;\mathrm{and}\;G_{23}$ represent the transverse elastic modulus, transverse Poisson’s ratio, and transverse shear modulus, respectively. These three transverse elastic constants are not completely independent and meet the relationship of isotropic materials:(3)$$G_{23}=E_{22}/\left(2+2v_{23}\right)$$

For the 3D woven composites, the braiding and the axial fibers are both considered as the matrix-impregnated fiber bundles, which are thought to be transversely is tropic. For the matrix-impregnated fiber bundles, the bridging matrix represents the correlation matrices of the stress increments in the constituent fiber and matrix, which is shown as follows:(4)$$\left\{d\delta_{i}^{m}\right\}=\left[A_{ij}\right]\left\{d\delta_{i}^{f}\right\}\;\;\;\left\{d\delta_{i}\right\}={\left\{d\delta_{11},d\delta_{22},d\delta_{33},d\delta_{23},d\delta_{12},d\delta_{13}\right\}}^{T}$$
$\left\{d\delta_{i}^{m}\right\}\mathrm{and}\left\{d\delta_{i}^{f}\right\}$ demonstrate the matrix of matrix stress increments and fiber stress increments, respectively; $\left[A_{ij}\right]$ represents the bridging matrix.
(5)$$\left[A_{ij}\right]=\left[\begin{array}{cccccc} a_{11} & a_{12} & a_{13} & 0 & 0 & 0\\ a_{21} & a_{22} & a_{23} & 0 & 0 & 0\\ a_{31} & a_{32} & a_{33} & 0 & 0 & 0\\ 0 & 0 & 0 & a_{44} & 0 & 0\\ 0 & 0 & 0 & 0 & a_{55} & 0\\ 0 & 0 & 0 & 0 & 0 & a_{66} \end{array}\right]\;\{\begin{array}{l} a_{11}=E^{m}/E_{11}^{f}\\ a_{22}=a_{33}=a_{44}=\beta +(1-\beta )\frac{E^{m}}{E_{22}^{f}}\;\;(0<\beta <1)\\ a_{12}=a_{13}=\frac{S_{12}^{f}-S_{12}^{m}}{S_{11}^{f}-S_{11}^{m}}(a_{11}-a_{22})=\frac{E_{11}^{f}v^{m}-E^{m}v_{12}^{f}}{E_{11}^{f}-E^{m}}(a_{22}-a_{11})\\ a_{21}=a_{31}=a_{23}=a_{32}=0\\ a_{55}=a_{66}=\alpha +(1-\alpha )\frac{G^{m}}{G_{12}^{f}}\;\;(0<\alpha <1) \end{array}$$

The bridging model does not set the geometry of feature voxel in advance. In theory, its volume can be infinitely small, and it does not impose boundary conditions and loads. Instead, it directly starts with the characteristic analysis and logical analysis of the bridging matrix connecting the mean stress of the matrix and the mean stress of the fiber, and then determines the elements of the bridging matrix by referring to the classical theoretical results.

In the above formulas, β and α are bridging parameters, which are respectively used to characterize the influence of fiber geometric characteristics on the transverse modulus $E_{22}$ and axial (in-plane shear) modulus $G_{12}$ of unidirectional composites. Generally, β=0.3∼0.6,α=0.3∼0.6. $S_{11}^{f},S_{12}^{f},S_{11}^{m},S_{12}^{m}$ are the flexibility matrix elements of fiber and matrix.

Moreover, the relationship between the bridging matrixing and the elastic constants of the matrix-impregnated fiber bundles can be presented as:(6)$$S=\left[\begin{array}{cccccc} \frac{1}{E_{L}} & -\frac{V_{LT}}{E_{T}} & -\frac{V_{LZ}}{E_{Z}} & 0 & 0 & 0\\ -\frac{V_{LT}}{E_{L}} & \frac{1}{E_{T}} & -\frac{V_{TZ}}{E_{Z}} & 0 & 0 & 0\\ -\frac{V_{LZ}}{E_{Z}} & -\frac{V_{TZ}}{E_{T}} & \frac{1}{E_{Z}} & 0 & 0 & 0\\ 0 & 0 & 0 & \frac{1}{G_{TZ}} & 0 & 0\\ 0 & 0 & 0 & 0 & \frac{1}{G_{LT}} & 0\\ 0 & 0 & 0 & 0 & 0 & \frac{1}{G_{LZ}} \end{array}\right]$$

Then, the mechanical properties of the matrix-impregnated fiber bundles can be calculated and shown in Table 2.

The 3D woven composite materials are continuous on a macro-scale, while they can be considered as spatial structures of constituent materials at the micro-level. Therefore, in the process of finite element analysis of 3D woven composite materials, the micro-unit cell model is adopted as a discrete element to perform macro-grid division of the three-dimensional braided composite material, and numerical analysis is carried out on the micro-unit cell model. For the convenience of calculation, the micro-unit cell model is defined as a rectangular element, which includes a rectangular matrix element of isotropic elastic material and a fiber element with uniaxial stiffness [42,43,44]. Firstly, the matrix element is defined as eight-node hexahedral, while the fiber element is treated as two-bar element. Secondly, due to the same shape matrix being applicable to the matrix element and fiber element, the each node’s displacement of the fiber element is determined by the node’s displacement of the matrix element. As shown in Figure 5, the degrees of freedom at node m and n are subjected by nodes A,B,C,D,E,F,G,H. Lastly, based on the transformation relationship between the local and global coordinate systems of the matrix and fiber, the stiffness matrix under the global coordinate system of the matrix and fiber is derived and superimposed [45].

The function of the matrix element’s displacement can be expressed as:(7)$$u=\sum_{i=1}^{8}N_{i}u_{i}\;v=\sum_{i=1}^{8}N_{i}v_{i}\;w=\sum_{i=1}^{8}N_{i}w_{i}\;$$
where $N_{i}$ is the matrix element’s shape function, which is an explicit function of the local coordinate system $\xi_{i}\;\eta_{i}\;\varsigma_{i}$.

The nodal displacement vector of the element can be expressed as:(8)$${\left\{\delta_{i}\right\}}^{e}={\left[u_{i},v_{i},w_{i}\right]}^{T}$$
where $u_{i},v_{i},w_{i}\left(i=1,2\ldots 8\right)$ represent the i node’s displacement along the x,y,z direction, respectively.
(9)$$\begin{array}{l} {\left\{\sigma \right\}}^{e}={\left\{\delta_{x}\;\delta_{y}\;\delta_{z}\;\tau_{yz}\;\tau_{zx}\;\tau_{xy}\right\}}^{T}=\left[D\right]\left[B\right]{\left[\delta \right]}^{e}=\left[D\right]\left[B_{1}\;B_{2}\;\ldots \;B_{8}\right]{\left[\delta_{1}\;\delta_{2}\;\ldots \;\delta_{8}\right]}^{T}\\ {\left\{\varepsilon \right\}}^{e}={\left\{\varepsilon_{x}\;\varepsilon_{y}\;\varepsilon_{z}\;\gamma_{yz}\;\gamma_{zx}\;\gamma_{xy}\right\}}^{T}=\left[B\right]{\left[\delta \right]}^{e} \end{array}$$

[B] represents the strain matrix and its block matrix is expressed as:(10)$$\left[B_{i}\right]=\left[\begin{array}{ccc} \frac{\partial N_{i}}{\partial x} & 0 & 0\\ 0 & \frac{\partial N_{i}}{\partial y} & 0\\ 0 & 0 & \frac{\partial N_{i}}{\partial z}\\ \frac{\partial N_{i}}{\partial y} & \frac{\partial N_{i}}{\partial x} & 0\\ 0 & \frac{\partial N_{i}}{\partial z} & \frac{\partial N_{i}}{\partial y}\\ \frac{\partial N_{i}}{\partial z} & 0 & \frac{\partial N_{i}}{\partial x} \end{array}\right]$$

[D] is the elastic matrix and it is shown as:(11)$$D=\frac{E^{m}(1-v)}{\left(1+v)(1-2v\right)}\left[\begin{array}{cccccc} 1 & \frac{v}{1-v} & \frac{v}{1-v} & & & \\ & 1 & \frac{v}{1-v} & & & \\ & & 1 & & & \\ & & & \frac{1-2v}{2(1-v)} & & \\ & & & & \frac{1-2v}{2(1-v)} & \\ & & & & & \frac{1-2v}{2(1-v)} \end{array}\right]$$
where $E^{m}\;\mathrm{and}\;v$ represent the elastic constants and passion ratio of matrix, respectively.

[J] is the three-dimensional Jacobian matrix, and its expression is:(12)$$\left[J\right]=\left[\begin{array}{cccc} \frac{\partial N_{1}}{\partial \xi } & \frac{\partial N_{2}}{\partial \xi } & \cdots & \frac{\partial N_{8}}{\partial \xi }\\ \frac{\partial N_{1}}{\partial \eta } & \frac{\partial N_{2}}{\partial \eta } & \cdots & \frac{\partial N_{8}}{\partial \eta }\\ \frac{\partial N_{1}}{\partial \varsigma } & \frac{\partial N_{2}}{\partial \varsigma } & \cdots & \frac{\partial N_{8}}{\partial \varsigma } \end{array}\right]\left[\begin{array}{ccc} x_{1} & y_{1} & z_{1}\\ x_{2} & y_{2} & z_{2}\\ \vdots & \vdots & \vdots \\ x_{8} & y_{8} & z_{8} \end{array}\right]$$

Based on the derived matrix of strain and elasticity, the stiffness matrix of the matrix element can be gained as:(13)$$\left[k_{m}\right]={∭\left[B\right]}^{T}\left[D\right]\left[B\right]dxdydz=\left[\begin{array}{cccc} k_{1,1} & k_{1,2} & \cdots & k_{1,8}\\ k_{2,1} & k_{2,2} & \cdots & k_{2,8}\\ \vdots & \vdots & & \vdots \\ k_{8,1} & k_{8,2} & \cdots & k_{8,8} \end{array}\right]$$

The expression for each submatrix can be listed as:(14)$$\left[k_{i,j}\right]=∭\left[B_{i}\right]\left[D\right]\left[B_{j}\right]dxdydz=∭\left[B_{i}\right]\left[D\right]\left[B_{j}\right]\left|J\right|d\xi d\eta d\zeta$$

For the stiffness matrix of the fiber element, the displacement mode of the fiber is consistent with matrix. Therefore,
(15)$$u=\sum_{i=1}^{8}N_{i}u_{i}\;v=\sum_{i=1}^{8}N_{i}v_{i}\;w=\sum_{i=1}^{8}N_{i}w_{i}\;$$

The nodal displacement vector of the fiber element is displayed as:(16)$${\left[\delta_{m}\;\delta_{n}\right]}^{T}={\left[u_{m},v_{m},w_{m},u_{n},v_{n},w_{n}\right]}^{T}$$
where $u_{m},v_{m},w_{m}\;$ and $u_{n},v_{n},w_{n}$ represent the displacement of node m and node n (as shown in Figure 5) along the direction of x,y,z, respectively.

To facilitate the calculation, fiber is defined to parallel to the x axis; hence, the transformation relationship between local coordinate system and fiber $x^{\prime }$ is expressed as:(17)$$\xi =f\left(x^{\prime }\right)\;\eta =f\left(x^{\prime }\right)\;\zeta =f\left(x^{\prime }\right)\;\left(-1\leq x^{\prime }\leq 1\right)$$

In the global coordinate system, the vector along the $x^{\prime }$ direction can be represented as:(18)$$r=\frac{\partial x}{\partial x^{\prime }}i+\frac{\partial y}{\partial x^{\prime }}j+\frac{\partial z}{\partial x^{\prime }}k$$

The cosine in the $x^{\prime }$ direction is:(19)$$\begin{array}{l} \cos\left(x^{\prime },x\right)=l_{1}=\frac{\partial x}{\partial x^{\prime }}/h\\ \cos\left(x^{\prime },y\right)=m_{1}=\frac{\partial y}{\partial x^{\prime }}/h\\ \cos\left(x^{\prime },z\right)=n_{1}=\frac{\partial z}{\partial x^{\prime }}/h\\ h=\sqrt{{\left(\frac{\partial x}{\partial x^{\prime }}\right)}^{2}+{\left(\frac{\partial y}{\partial x^{\prime }}\right)}^{2}+{\left(\frac{\partial z}{\partial x^{\prime }}\right)}^{2}} \end{array}$$

Because the fiber lies in $x^{\prime }$, axial stress is merely considered in the local coordinate system.

Therefore:(20)$$\begin{array}{l} \varepsilon_{p}=\frac{\partial u^{\prime }}{\partial x^{\prime }}=\left[l_{1}^{2}\;m_{1}^{2}\;n_{1}^{2}\;l_{1}m_{1}\;m_{1}n_{1}\;l_{1}n_{1}\right]\left[\varepsilon \right]=\left[T\right]\left[B\right]{\left[\delta \right]}^{e}=\left[B^{\prime }\right]{\left[\delta \right]}^{e}\\ \sigma_{p}=E_{f}\left[B^{\prime }\right]{\left[\delta \right]}^{e}\\ \left[K_{ij}\right]=\int_{-1}^{1}{\left[{B^{\prime }}_{i}\right]}^{T}E_{f}\left[{B^{\prime }}_{j}\right]Adl=\int_{-1}^{1}{\left[{B^{\prime }}_{i}\right]}^{T}E_{f}\left[{B^{\prime }}_{j}\right]Ahdx^{\prime } \end{array}$$

The stiffener element is composed of the matrix and fiber elements, so the stiffness matrix of the stiffener element is the superposition of the fiber stiffness matrix and the matrix stiffness matrix, shown as:(21)$${\left[K\right]}^{e}=\left[k_{m}\right]+\sum_{i=1}^{n}\left[k_{i}^{f}\right]$$
where n is the number of fibers, $\left[k_{i}^{f}\right]$ is the stiffness matrix of the nth fiber element.

### 3.3. The Method of Multi-Scale Finite Element

Generally, three-dimensional braided composite materials can be analyzed through three scales: micro-scale, meso-scale, and macro-scale, which are corresponding to the fiber bundles, the representative volume element of woven structure, and the integral fabric [46]. Firstly, the micro-scale RVE model is calculated to obtain the mechanical properties of fiber bundles based on the elastic properties of the fiber, matrix, and interface. Secondly, assign the results of the previous calculation to the material properties of the fiber bundle, and the mechanical properties of the matrix and interface remain coherently [47]. Similarly, the mechanical properties of three meso-scale RVE models can be obtained. Lastly, the homogenized results from the three meso-scale RVE models are substituted into the macro-scale to generate the overall mechanical performance. The entire analysis relies on Abaqus/standard. Figure 6 presents the procedure of finite element analysis of the 3D braided meniscus’s multi-scale model [48]. The fundamental step is to acquire the mechanical properties of matrix-impregnated fiber bundles composed of matrix and fiber. To facilitate for applying periodic boundary conditions, the hexagonal RVCs are used [49,50,51]. The predictive mechanical properties of fiber bundles and three RVCs based on multi-scale finite element method are listed in Table 3.

### 3.4. The Comparison of Artificial and Pure Matrix Meniscus

Taking an adult weighing 60 kg as an example, a load of 400 N, 600 N, 800 N, 1200 N, and 1400 N was applied to the meniscus to simulate the stress state of the knee joint meniscus during walking, running, jumping, and high-altitude falls. Import the established fiber-reinforced meniscus model into Abaqus, define the material properties and orientation, fix the bottom platform of the model, and apply pressure to the wedge-shaped surface for finite element analysis.

## 4. Results and Discussion

The designed structure of artificial meniscus shown in Figure 1 can excellently mimic the internal fiber distribution of the native meniscus. The designed specific production process flow and braiding steps can provide the feasibility of the preparation of artificial meniscus prosthesis. Besides, the innovative weaving structure of meniscus provides a new approach for the preparation of new biomimetic meniscus.

This paper adopted fiber-embedded matrix method and multi-scale finite element method to predict the elastic constants of the artificial meniscus model. The calculated results are listed in Table 2 and Table 3. By comparison, the mechanical properties of two models are basically consistent, which verified the rationality of modeling and accuracy of calculation results. The results apparently demonstrate the excellent strengthen ability of the fiber, and the direction of strength enhancement is highly consistent with the paths of fiber bundles. For example, the interior RVC have the fibers with vertical orientation, the elastic modulus 2.19 GPa in the vertical direction is significantly superior to the elastic modulus 0.2 GPa of the top RVC and 0.197 GPa of the bottom RVC. Moreover, the solved data provide a basis for further analysis of the macro-scale model of the meniscus.

Figure 7 shows the stress–vertical deformation curve of the 3D braiding meniscus and the pure matrix meniscus. It can be concluded the three-dimensional braided structure has an improved resistance to deformation by about 60% compared to the pure matrix support in the vertical direction, which indicates that the designed structure can significantly improve the previous characteristics of meniscus delamination and fracture.

## 5. Conclusions

The main objective of this paper is aimed at designing a new artificial braided meniscus model and predicted the elastic constants through the finite element analysis. we adopted the models of fiber-embedded matrix and multi-scale methods separately for finite element analysis to achieve the reliable elastic properties. Meanwhile, we compared the results for two models, which are basically consistent and verified the accuracy of analysis. Finally, we conducted the comparative simulation analysis of the meniscus model and the pure matrix meniscus model based on the solved elastic constants through Abaqus, as shown in Figure 8, which indicated a 60% increase in strength.

For future work, we will do more research on the bioinspired design and the semiinterpenetrating polymer network (semi-IPN). Besides, we will provide physical objects for experimental testing.

## Figures and Tables

**Figure 1 materials-16-04775-f001:**
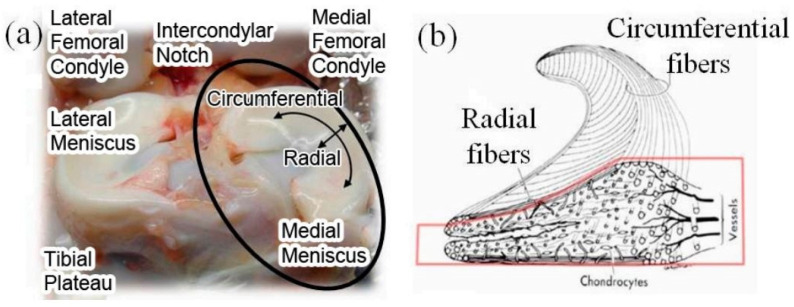
(**a**) Anatomical photograph of human meniscus; (**b**) internal microstructure at cross-section [26,27].

**Figure 2 materials-16-04775-f002:**
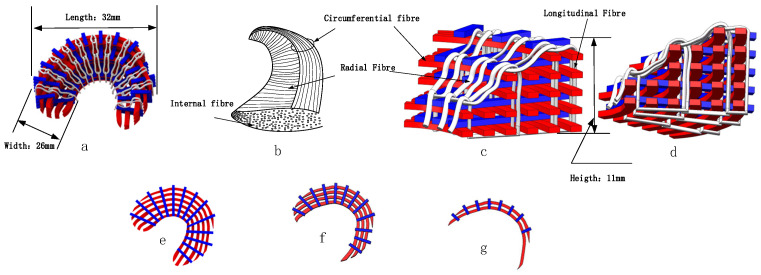
The designed structure of meniscus. (**a**) 3D braided structure of meniscus; (**b**) fiber distribution of physiological meniscus; (**c**,**d**) partial enlarged view of 3D braided meniscus; (**e**) the first level of the braided structure; (**f**) the second level of the braided structure; (**g**) the third level of the braided structure.

**Figure 3 materials-16-04775-f003:**
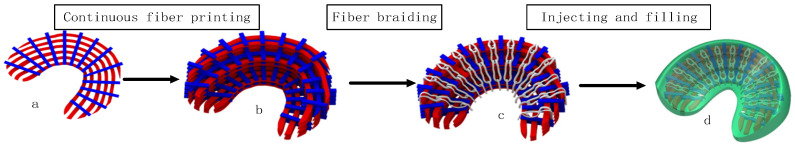
The 3D braiding technological process. (**a**) the first level of the braided structure; (**b**) the fiber distribution in each layer of the woven structure; (**c**) the intact woven structure; (**d**) the woven structure after injection molding.

**Figure 4 materials-16-04775-f004:**
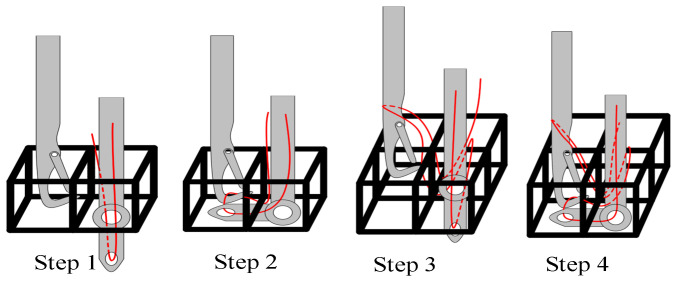
Detailed weaving steps.

**Figure 5 materials-16-04775-f005:**
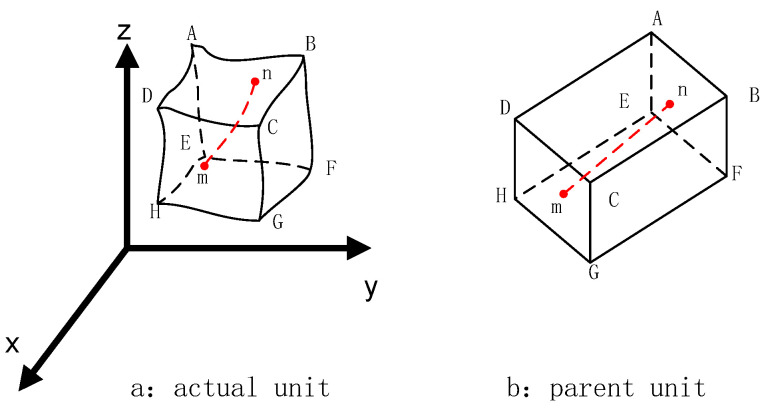
Unit cell model. (**a**) actual unit; (**b**) parent unit.

**Figure 6 materials-16-04775-f006:**
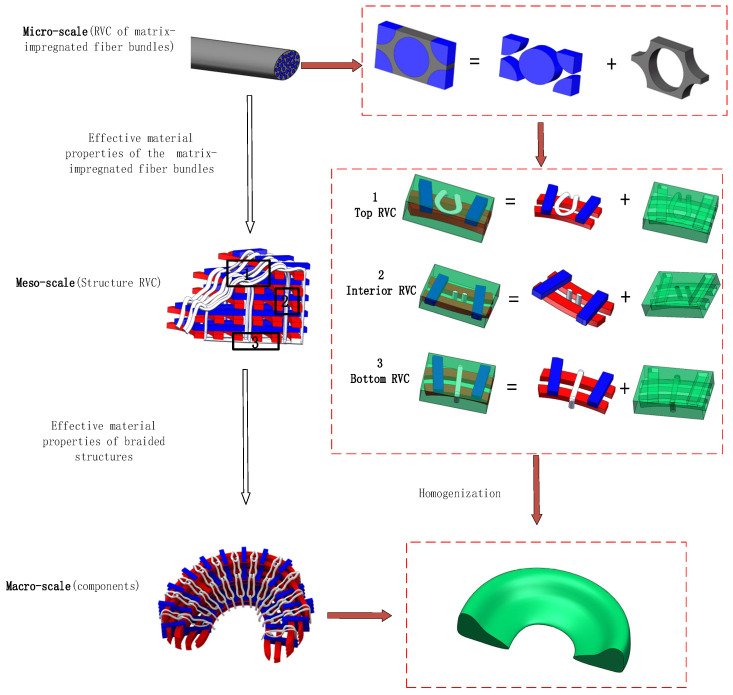
Schematic diagram of multi-scale analysis.

**Figure 7 materials-16-04775-f007:**
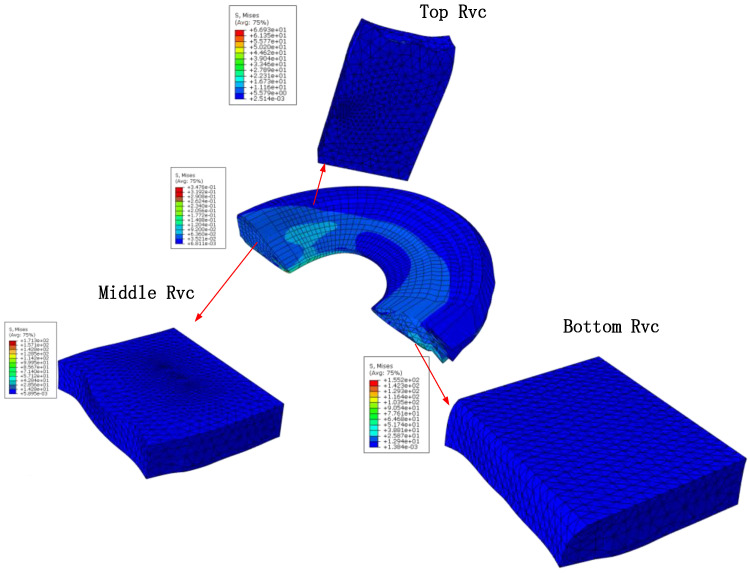
Predicted stress field contours.

**Figure 8 materials-16-04775-f008:**
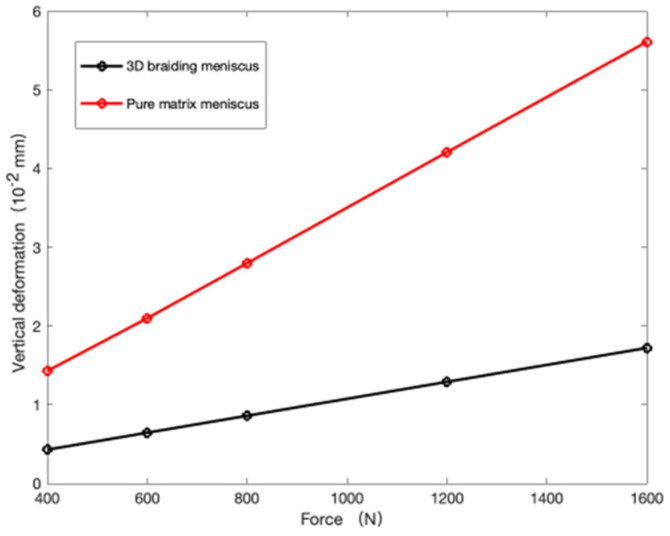
The stress–vertical deformation curve of the 3D braiding meniscus and the pure matrix meniscus.

**Table 1 materials-16-04775-t001:** The mechanical properties of Kevlar and PCU [33,39].

	Kevlar	Polycarbonate Polyurethane (PCU)
Longitudinal modulus (GPa)	128.7	0.057
Transverse modulus (GPa)	12.87	
Longitudinal passion ratio	0.3	0.43
Longitudinal shear modulus (GPa)	12.87	
Transverse shear modulus (GPa)	12.87	

**Table 2 materials-16-04775-t002:** The predictive mechanical properties of fiber bundles and three RVCs using the finite element method.

	$E_{L}$ (Gpa)	$E_{T}$ (Gpa)	$E_{Z}$ (Gpa)	$v_{LT}$	$G_{LT}$ (Gpa)	$G_{LZ}$ (Gpa)	$G_{TZ}$ (Gpa)
Fiber bundles	90.1	0.479	0.479	0.37	0.17	0.17	0.17
Top RVC	13.09	0.2	10.0	0.61	0.03	0.036	0.03
Interior RVC	11.44	2.19	10.76	0.63	0.03	0.039	0.03
Bottom RVC	13.6	0.2	9.09	0.67	0.03	0.037	0.03

**Table 3 materials-16-04775-t003:** The predictive mechanical properties of fiber bundles and three RVCs using the multi-scale finite element method.

	$E_{L}$ (Gpa)	$E_{T}$ (Gpa)	$E_{Z}$ (Gpa)	$v_{LT}$	$G_{LT}$ (Gpa)	$G_{LZ}$ (Gpa)	$G_{TZ}$ (Gpa)
Fiber bundles	91	0.417	0.417	0.31	0.15	0.15	0.15
Top RVC	13.23	0.2	10.12	0.67	0.03	0.035	0.03
Interior	11.44	2.19	10.76	0.63	0.03	0.039	0.03
Bottom	13.73	0.197	9.19	0.67	0.03	0.035	0.03

## Data Availability

Not applicable.

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
