# Peer review of "Design and Finite Element Analysis of Artificial Braided Meniscus Model"

_materials, 2023, doi:10.3390/ma16134775_

Round 1

Reviewer 1 Report

Review on the article Design and finite element analysis of artificial braided meniscus model, submitted to Materials.

It is difficult to clarify, which part of the manuscript is authors’ own study and which is described on the basis of the literature, especially as the aim of the study was not clearly stated. Currently, the manuscript looks more like a review rather than an article presenting authors’ own results. Moreover, the manuscript contains some typos (i.a. the lack of space between unit and its value, “Gpa” instead of GPa, etc.) as well as the lack of equations numbering, which lower its scientific soundness. Additional comments to each section are presented below. The authors should include appropriate changes to make their manuscript publishable in Materials.

Abstract

Abstract is missing clarification what was actually done by the authors – there is no real separation between current state of the art and authors’ contribution. Moreover, it is missing more detailed description of the results obtained in the research as well as appropriate conclusions.

1.      Introduction

The reference to similar studies is missing. It seems like the authors refer only to experimental studies (references 10-14) while conducting numerical research. If similar studies do not exist in the currently published literature, it should be appropriately described.

Description of the aim of the study is missing.

2.      Design of the meniscus’s 3d braiding structure and technology

General dimensions of the designed structured should be presented in Figure 1.

Figure 2 caption should contain description of labels a-d.

3.      Analytical method

Table 1 is missing references.

It is unclear what boundary conditions were assumed to obtain stresses presented in Figure 6.

The style of English language is fine, however typos mentioned in the comments and suggestions section are lowering its quality.

Reviewer 2 Report

The work presents an interesting approach in obtaining meniscus substitutes.

Anyway, great efforts should be made to improve the quality of the work.

Firstly, the authors should improve the introduction section by including some details on more recent published works on advanced technological approaches for customised and biomimetic structures for meniscal substitution (i.e., meniscus regeneration via precision 3D printing technologies).

In the work there is no mention on the possibility to design customized structures starting from images of natural menisci, from human or animal origin. The selected dimension for meniscal substitution refers to historic publication on degenerated menisci as a consequence of different pathologies.

The improvements related to the selected design criteria and the advantages of the proposed methodology should be better described. The proposed structures should be also better described in terms of shape and dimension, i.e., of the single filament (of both the 3D braided or 3D printed portion of the artificial meniscus), considered as analogues of the collagen fibres distributed in the natural meniscus.

The possibility to provide a more marked bioinspired design for meniscal substitutes should be also taked into account by including injectable systems based on, in example, semiinterpenetrating polymer network (semi-IPN) obtained by promoting collagen fibrillogenesis in the presence of hyaluronic acid (HA).

Furthermore, the possibility to provide a combination design of meniscal substitutes should be also considered: predictive mechanical properties may provide some issues on analytical formulations that properly described the uni- or multi-axial stress–strain behavior.

Images of the proposed meniscus substitutes should be provided, also giving detailes (i.e., SEM images) of their morphology. Further studies are required because of the lack of correlation between obtained results and preparation method which clearly affects the morphology, surface topography, and, consequently, the performances of proposed structures. The present work should provide an initial step towards future research with the aim of finding a complex correlation, which will also involve the effect of the preparation method.

References should be updated.

Reviewer 3 Report

Dear authors,

please find some comments on your document in order to improve it:

-if latin terms are included in the document, please use italics font for them

-"polyvinyl alcohol": poly(vinyl alcohol) and "polylactic acid": poly(lactic acid) are the proper writing 

-"fatigue, and density": never use a comma (,) before the words "and" or "or" when similar things are in parathesis. Check and correct throughout the text

-"(CFRTPCs)": you may omit it

-et al. in italics in every case

-always keep a space between the numbers and the units, check and correct where needed in text

-You may enrich the Introduction with some extra theory on the implication of 3D-printing for biomaterials

-use a dash after 3D, pe. "3D-braiding". check and correct throughout the text

-l.70-74: use semicolons (;) with lowercase letters of fulstop (.) for capital letter

-use a space figure 1a, figure 1b etc

-use a dash " half-moon"

-"Polycarbonate polyurethane": there is no such term or trade name. Write the chemical structure of the copolymer or the ingredients of the composite material so to identify the correct name.

-4. Discussion

-Table 2, Table 3

-GPa the right unit

If language enrichment is possible, please do so...

Round 2

Reviewer 1 Report

Review on the revised version of the article Design and finite element analysis of artificial braided meniscus model, submitted to Materials.

The authors conducted major changes in the manuscript on the basis of critical comments presented within previously done review process. These changes greatly increased article's quality and for this reason my suggestion is to accept this paper for publishing within Materials.

However, the authors consistently made a typo within the article, which should be changed while making last changes prior to publication. Unit Gigapascal is written "GPa" and not "Gpa". 

Congratulations on the study and best of luck in conducting more research.

There are no major issues detected in the quality of English Language that would decrease scientific soundness of the article.

Reviewer 2 Report

The authors did not fully respond to the suggestions of the first revision stage.

The introduction has been revised by the authors, but it does not adequately outline the area of investigation. Furthermore, in my opinion, throughout the manuscript there seems to be a lack of real connection between hypotheses and results.and results.

Reviewer 3 Report

MPa and GPa are the right units, they were correct